# Development and Preliminary Testing of Porcine Blood-Derived Endothelial-like Cells for Vascular Tissue Engineering Applications: Protocol Optimisation and Seeding of Decellularised Human Saphenous Veins

**DOI:** 10.3390/ijms23126633

**Published:** 2022-06-14

**Authors:** Andrew Bond, Vito Bruno, Jason Johnson, Sarah George, Raimondo Ascione

**Affiliations:** Translational Health Sciences, Bristol Medical School, University of Bristol, Bristol Royal Infirmary, Bristol BS2 8HW, UK; andrew.bond@bristol.ac.uk (A.B.); vito.d.bruno@bristol.ac.uk (V.B.); jason.l.johnson@bristol.ac.uk (J.J.); s.j.george@bristol.ac.uk (S.G.)

**Keywords:** endothelial colony forming cells, cell seeding, endothelium, bioengineering, vascular graft

## Abstract

Functional endothelial cells (EC) are a critical interface between blood vessels and the thrombogenic flowing blood. Disruption of this layer can lead to early thrombosis, inflammation, vessel restenosis, and, following coronary (CABG) or peripheral (PABG) artery bypass graft surgery, vein graft failure. Blood-derived ECs have shown potential for vascular tissue engineering applications. Here, we show the development and preliminary testing of a method for deriving porcine endothelial-like cells from blood obtained under clinical conditions for use in translational research. The derived cells show cobblestone morphology and expression of EC markers, similar to those seen in isolated porcine aortic ECs (PAEC), and when exposed to increasing shear stress, they remain viable and show mRNA expression of EC markers similar to PAEC. In addition, we confirm the feasibility of seeding endothelial-like cells onto a decellularised human vein scaffold with approximately 90% lumen coverage at lower passages, and show that increasing cell passage results in reduced endothelial coverage.

## 1. Introduction

Autologous saphenous veins (SV) are the most commonly used conduits to bypass blockages in the coronary and peripheral arteries of affected patients thanks to their long length and lack of atherosclerosis. One of the biggest issues facing patients following coronary (CABG) and peripheral (PABG) artery bypass grafting surgery using SV is the high incidence of early thrombosis and late restenosis. Indeed, following CABG surgery 15–20% of SV grafts (SVG) suffer early thrombosis within one year, and only 50% remain patent ten years post-surgery due to intimal thickening [1]. The rates of these life-threatening complications are worse following PABG surgery: 20% lower extremity graft failure within one year and 50% failure by year five [2,3].

Endothelial cells (ECs), which line all blood vessels, are the first line of defence between the vessel wall and the thrombogenic blood; disruption of this layer can lead to early thrombosis, inflammation, and subsequent vessel restenosis. The advent of protocols such as the no-touch technique, where the SV is removed with a pedicle of surrounding tissue [4,5], have been shown to better preserve the endothelium compared to conventional techniques [6]. However, the process of SV harvesting, including physiological pressure distension to check for leaks, and their ischemic time before being used for grafting and allowed on perfusion, can itself cause damage and denudation of the ECs [7]. Immediately following venous harvest the proportion of viable EC is reduced to around 70%, and 24 h of culture ex vivo results in further decline to 66% [8]. However, it has been shown that preservative solutions play a large part in EC viability, with reports of 0–25% viability within 90 min of harvest and up to 76–100% viability after 24 h, depending on storage conditions [9].

An avenue for improving long term graft patency is therefore tissue engineering of optimal vascular conduits able to enhance and protect EC coverage before and possibly after surgical implantation. A key feature of CABG and PABG is to use autologous SV to both prevent immunosuppression and to ensure that the internal lining of autologous ECs prevents thrombosis. Hence, adopting these key factors may be critical in modern vascular tissue engineering. Techniques to isolate and culture autologous EC from patients listed for CABG/PABG surgery are not a viable option. An alternative approach is to use ECs derived from autologous blood, called endothelial colony forming cells (ECFC). ECFCs, also known as blood outgrowth endothelial cells (BOEC) or late outgrowth endothelial progenitor cells, have attracted attention in this field. They have potential for autologous cell therapy for vascular repair, gene therapy, investigation of EC disease, and tissue bioengineering thanks to their ability to differentiate into mature ECs and promote vascular formation [10,11]. ECFCs have already been used on graft conduits in different species [12,13,14,15,16], have been shown to rapidly proliferate, reaching 10^19^ cells within six weeks of continuous culture [17], and maintain a stable phenotype through a 1 × 10^9^-fold expansion [18].

Protocols for deriving ECFC from human blood are readily available [19,20,21], however only a small number of studies have reported deriving endothelial-like cells (ELC) from porcine blood [22,23,24,25,26,27,28,29], and at no time have these been seeded directly onto the luminal surface of a tissue-engineered vascular graft. In one study, ECFCs were seeded into the adventitia at a vein-to-PTFE graft anastomosis, with the suggestion of reduced intimal hyperplasia [27].

The aim of this study was to develop a replicable isolation protocol for porcine blood-derived endothelial-like cells and then use these for luminal seeding into a human decellularised vascular graft.

## 2. Results

### 2.1. ELC Isolation Success

For all blood collections, large cell pellets of peripheral blood mononuclear cells (PBMC) were obtained; following 24 h in culture, large numbers of cells (monocytes) had stuck down. The presence of colonies of cells with a cobblestone-like morphology, indicative of successful EC differentiation, was low. PBMCs from 20% of pigs sampled (7 out of 35) and 18% of individual batches processed (10 out of 57) successfully differentiated into cobblestone-forming colonies. The first colonies appeared on average after 10.3 ± 2.1 days, although in six cases colonies appeared earlier, between days 7 and 9 (mean 8.2 ± 0.4 days), while the remaining experiments showed PBMCs becoming ELCs after 23 days. PBMCs from approximately half of the pigs (49%, 17 out of 25) and 44% of individual batches (25 out of 57) did not differentiate into ELCs, despite large numbers of adherent monocytes being present in the first few days. PBMCs from 12 of the pigs (34%) and 22 batches (39%) differentiated into a different type of cell; rather than cobblestone colonies, they grew resembling a spiderweb, with very spindly cells that formed connections with neighbouring cells until they reached confluence. These cells, which proliferated rapidly, are referred to here as blood outgrowth cells (BOCs). PBMCs in separate culture wells from one pig went on to differentiate into both ELCs and BOCs, depending on the culture conditions they were exposed to. PBMC from blood obtained from commercial pigs processed at the abattoir without GA went on to develop ‘spiderweb-like’ BOCs, with the presence of BOCs being observed on average after 7.5 ± 0.7 days (range 6–14 days).

The various conditions attempted for isolation of ELCs are summarised in Table 1 and Table 2. The protocol that yielded the most success involved the following factors: blood collected from donor pigs with no previous procedure, subjected to short period of GA, processed immediately, exposed to long centrifugation time (20 min), and with isolated PBMCs cultured in Lonza endothelial growth media-2 (EGM-2; Cambridge, UK) media.

In addition, removal of RBCs using lysis buffer slightly improved isolation success; however, not removing residual RBC was not a prerequisite for successful isolation.

Withdrawing blood from control pigs that had undergone a previous procedure appeared to be the worst-case scenario, with 0/17 successful ELC isolations compared to 6/26 (23%) successful ELC isolations in donor pigs that had not been used before.

### 2.2. Cell Characterisation

#### 2.2.1. Immunocytochemistry

Cells that had a cobblestone morphology grew in a monolayer, were shown by immunocytochemistry to be positive for the endothelial markers platelet EC adhesion molecule (PECAM-1; CD31), vascular endothelial (VE)-Cadherin, von Willebrand factor (vWF), and Dolichos Biflorus Agglutinin (DBA)-Lectin, and were positive for vimentin (Figure 1). The staining pattern of CD31 and VE-Cadherin was similar between ELC and porcine aortic EC (PAECs), with the proteins being localised at the cell borders. However, there was much greater expression of vWF and lower expression of vimentin in ELCs compared to PAECs. All ELCs and PAECs were negative for smooth muscle myosin heavy chain (SM-MHC), alpha-smooth muscle actin (SMA), CD90, and CD45 (images not shown).

Cells that did not have a cobblestone morphology, i.e., blood outgrowth cells (BOC), grew at a higher proliferative rate, tended to grow in multiple layers, were positive for CD90, vimentin, and alpha-SMA, contained small patches of cells expressing vWF, DBA-lectin, and VE-Cadherin, and were negative for CD31, SM-MHC, and CD45 (Figure 2).

#### 2.2.2. Exposure to Flow Conditions

mRNA expression data are shown as the fold change from PAECs under static conditions (Table 3) or using ΔCt values (Figure 3). All ELCs remained attached to the plates for the duration of the shear stress period, enabling RNA to be collected.

Compared to aortic ECs, ELCs had higher expression levels of CD31, VE-Cadherin and vWF, while increasing shear stress had no effect of. eNOS expression did not change between cell types or when differing shear stress levels were applied.

### 2.3. ELC Seeding/Endothelial Coverage

Seeding the luminal surface of decellularised human saphenous vein with porcine ELCs resulted in successful endothelial coverage of 68.9 ± 8.8% (range from 23% to 97%) (Figure 4a,c), and showed continued expression of CD31 (Figure 4d). Unseeded controls showed no staining by DBA-Lectin (Figure 4f). Increasing the passage of the seeded ELCs appeared to negatively impact seeding success with cells at P6 showing the highest success (91.2 ± 3.9%), while cells at only P9 successfully covered 28.3% of the lumen (Figure 4b).

## 3. Discussion

In this paper, we have described the various approaches taken to deriving endothelial-like cells from pig blood, allowing us to refine and develop an effective protocol for porcine ELC isolation for future use in translational research studies. In order to prevent unnecessary use of additional pigs and to comply with the principles of the three Rs (Replacement, Reduction, and Refinement of animals in research), blood samples were collected at termination from control cases of multiple studies being undertaken in the Translational Biomedical Research Centre. However, in all these cases the same breed of pig was used, along with similar weight and age range. In this study we were unable to definitively characterise the isolated cells as ECFC, mature EC, circulating EC, or endothelial progenitor cells; hence, the generic term endothelial-like cells is adopted. The isolated cells are presumed to be ECFCs based on the protocols followed, and the ability of the cells to maintain the EC markers CD31, VE-Cadherin, vWF, eNOS, and DBA-lectin for multiple passages suggests they are appropriate for use in translational tissue engineering studies. Although assessment of clonal ability was not performed, it was consistently observed that the cells formed colonies when passaged and plated (Figure A1). However, to confirm that cells are ECFC, further studies would be required to investigate key determinants of ECFC such as clonogenic ability or angiogenic capacity as has been previously described [10,19,30]. Although for clinical purposes the nomenclature is perhaps not as relevant as the practical benefit, accurate characterization and nomenclature has been raised as an issue of great importance by other researchers [10]. However, if the isolated cells express EC markers and line the lumen of seeded vessels as a monolayer, then they have clinical translational potential, and as such, should be pursued further.

We successfully isolated ELCs from 20% of pigs sampled. However, we appreciate that by applying the criteria ascertained in this study it might be possible to markedly increase the success rate in future studies. Deriving ELCs from human blood has been associated with previous success rates ranging from 40–60% [31]. Previous studies in humans have shown there are only a limited number of circulating monocytes with the potential to become ECs, and in pigs it has been estimated that only 5 ± 2 ECFC colonies form per 100 mL blood [24]. In the same study, it was ascertained that the number of ECFC colonies increases three-fold immediately post-MI, then returns to basal levels seven days later. Estimates for different blood-derived endothelial cells calculate there to be 2.6 ± 1.6 circulating endothelial cells/mL [32], 1 ECFC per 10^6^–10^8^ PBMCs (0.001–0.000001%) [33], and circulating ECs or endothelial progenitor cells are thought to represent only 0.01% to 0.0001% of PBMCs [34]. Reference values for healthy pigs (data provided by Langford Diagnostic Labs, Bristol, UK) are 4.6–10 × 10^6^/mL (lymphocytes) and 0.3 × 10^6^–1.2 × 10^6^/mL (monocytes). Taking the median values of each (7.3 × 10^6^/mL and 0.75 × 10^6^/mL lymphocytes and monocytes, respectively) would assume approximately 8 × 10^6^ PBMC/mL. Estimates for the number of pig ELCs based on the pig and human data above would indicate potential ranges of 0.05 cells/mL and 0.08–800 cells/mL of blood respectively. This suggests that there should be large numbers of cells available; yet, we only managed to isolate from 20% of animals, possibly due to the negative factors identified in this study. For example, collecting blood from pigs that had undergone a previous procedure triggered 0/17 successful ECFC isolations. On the positive side, all successful isolations used supplemented EGM-2 media (Lonza, Cambridge, UK) as opposed to endothelial cell growth medium-2 (ECGM-2) media (Promocell, Heidelberg, Germany). Both media use the same named supplements; however, the concentration in EGM-2 is proprietary information, and until this information is made available it is unclear why this media appears superior for ECFC growth.

Due to the presumably small numbers of available progenitor cells, we believed that in order to optimise our chances of isolating ELCs from pigs large volumes of blood should be collected. As 350 mL of blood could be processed in one batch, the blood/DPBS suspensions were layered onto the Ficoll–Paque density separation solution in 40 Sepmate-50 tubes. Despite Ficoll–Paque not being thought to be toxic to PBMCs for short durations, increasing the time between layering the blood and centrifuging the tubes can result in increased mixing at the blood/solution interface. This may have inadvertently reduced the percentage of mononuclear cells recovered.

Increased red blood cells in the isolated cell pellet may have a negative effect on the process of isolating ELCs. Any RBCs that were transferred to the culture plates with the mononuclear cells, may have rapidly undergone eryptosis (programmed cell death) and triggered signalling cascades that would normally result in their phagocytosis from the circulation by the immune system [35,36]. Haematocrit levels (the ratio of the volume of RBC to the total volume of blood) in pigs and humans are very similar (36–43% pigs [37] and 36–53% in humans [38]); however, Sepmate-50 tubes (StemCell Technologies) have been designed and optimised for human blood use, and as such, red blood cells remained present in the pellet. Baseline haematology data of blood taken under anaesthesia from eleven pigs enrolled in another study had a mean haematocrit level of 30.4 ± 0.6%, below ‘normal’ reference ranges, and this may be a factor in increased RBCs in the pellet. Following the manufacturer’s instructions, the centrifugation time was increased; while this did reduce the RBC content, the reduction did not reach a level that warranted dispensing with the lysis buffer. However, lysis buffer did not appear to reduce the likelihood of successful isolation.

It is of interest that it was not possible to derive ELCs from any pigs that had undergone a previous procedure. The majority of previous procedures carried out were part of an induced percutaneous myocardial infarction model with temporary intra-coronary balloon inflation. There is now evidence suggesting that tissue damage and acute myocardial infarction causes the mobilisation of endothelial progenitor cells from the bone marrow into the circulation [39,40,41]. If all of the cells honed into the site of injury, then it may have been that at termination when blood was collected (typically four weeks post-MI) there were fewer cells left in the circulation for derivation into ELCs. This is an intriguing speculation that warrants additional probing in future studies.

An alternative reason for the reduced success if isolation is that pigs that have been subjected to previous procedures were under GA for a long period due to MRI imaging acquisition before blood withdrawal, which may have had a detrimental effect on PBMC health. It has been shown that local and general anaesthetics commonly used in and immediately prior to surgery reduce proliferation of PBMCs [42], and following elective cardiac surgery, monocyte plasticity (where monocytes become dendritic cells) can be lost for up to three months [43]. Extrapolating these findings to our ELC isolations could suggest that the PBMCs and their endothelial progenitors were rendered defective by prior procedures and prolonged GA. Additionally, there have been studies suggesting that magnetic fields such as those in MRI scanners have effects on peripheral blood immune cells and lymphocyte DNA integrity [44,45,46]. This should be taken into consideration in future studies involving only donor pigs subjected to clinical-grade blood collection under short GA times.

In addition, we observed that isolation from slaughtered commercial pigs without GA was not associated with successful ELC isolation, despite the blood being promptly collected into either ethylenediaminetetraacetic acid (EDTA) or heparin, which is not uncommon for human ELC isolations. However, all of these samples had a longer time from blood collection to processing, which might explain the lack of success. All PBMCs and subsequent adherent monocytes went on to differentiate into very rapidly growing blood outgrowth cells. Another explanation for lack of success could be the method of slaughter and blood collection, which might have triggered fibroblast contamination from the stick wound; however, all samples were collected after blood had been flowing for a few seconds to flush the wound, and this is therefore unlikely. Had this approach been successful for ELC isolation it would have reduced the need for porcine “clinical grade” blood from the termination of other studies, which would represent a more readily available and accessible supply of blood and reduce associated costs. Instead, further investigation needs to be carried out in order to determine exactly what the BOCs are as well as to determine whether they are of use in future tissue engineering studies.

We were able to characterise isolated ELCs by ICC staining and mRNA expression and to compare them to isolated PAECs. ELC and PAEC show similar staining patterns for the EC markers CD31 (PECAM-1) and VE-Cadherin, with protein expression on the cell periphery at the cell–cell junction. ELC had increased expression of CD31 and VE-Cadherin mRNA under static and shear stress induced conditions, suggesting decreased permeability of these cells [47]. VE-Cadherin and CD31 are crucial complexes for sensing fluid shear stress and transducing tensile forces through the endothelium [48,49,50]; thus, an increase in expression may be of benefit when seeding a vascular graft.

vWF staining was higher in ELCs than PAECs, and mRNA expression confirmed this finding. There was a large amount of variability between mRNA expression in ELC, as highlighted by the large SEMs. This finding is not novel, and is in keeping with previous studies; it is thought to be a reflection of maximum cell density [51], or possibly due to heterogeneity in the blood-derived EC population. Future studies might aim to separate the populations according to clonal ability in order to provide a more homogenous population, which is something that may need to be taken into consideration if cells are used for seeding vein grafts. While vWF has multiple roles, its predominant role is in thrombosis and haemostasis; it has roles in angiogenesis and smooth muscle cell proliferation as well, and the circulatory level of vWF has been reported to increase in presence of inflammation [52]. It is not clear whether increased levels of vWF trigger blood clotting and thrombosis in vein grafts; however, in the clinical setting, anticoagulation therapy takes place in the presence of inflammation in order to prevent deep vein thrombosis.

There were no obvious differences in mRNA expression of eNOS between PAEC and ELC. Typically, eNOS is increased under laminar shear stress flow conditions [53,54]; however, that was not seen in either cell type in the present study. This could be due to the orbital shaker method used here, which generates a range of shear stresses within the plate [55] that could blunt the observable response.

Being able to isolate ELC from pigs could be of great research value for future studies testing the seeding potential of these cells on human decellularised venous grafts, or indeed on other bioengineered vascular conduits ex vivo or in porcine models without the need for immunosuppression [56]. Ultimately, this approach could lead to studies in patients aimed at using patients’ own blood to isolate ELCs for seeding onto their own decellularised vein or on bioengineered vascular scaffolds before surgical implantation. 

In this study, we have confirmed the feasibility of seeding the isolated porcine ELCs onto a decellularised human vein scaffold using a method where we tied the cells into the lumen, seeded them on a roller at very low speed, and then cultured them under static conditions to enable increased lumen coverage up to 97%, which grow as a monolayer similar to that seen with ECs. Increasing the passage of cells reduced the success of cell seeding and decreased the percentage of lumen covered by the cells (Figure 4b), suggesting that early-passage cells should be used for this purpose. This is not in keeping with results from human studies, which suggest that blood endothelial cells can be used up to very high numbers and from later passages without impacting cell proliferation [18]. We are currently planning future studies involving the implantation of D-hSVs seeded with porcine ELC in a porcine model of carotid artery replacement, with the aim of reducing vein graft thrombosis at 4–12 weeks.

### Blood Outgrowth Cells (BOCs)

Non-cobblestone colony forming cells (BOCs) were derived from porcine PBMCs. Of note, cells that did not develop a cobblestone morphology were much easier to remove during enzymatic dissociation with TryplE Express, requiring only 30–90 s to detach compared to approximately 10 min for ELCs. Experiments to subject the BOCs to shear stress on the orbital shaker were attempted, however, even when subjected to low shear stresses the cells had all detached before the end of the experiment duration. These cells stained strongly positive for vimentin and CD90, which are markers of fibroblasts and cells of mesenchymal origin, and the majority of cells were positive for alpha-smooth muscle actin, which is a marker of both smooth muscle cells and activated fibroblasts. On the other hand, they were negative for CD31 and VE-Cadherin, suggesting that they are not endothelial-like cells; however, there were patches of cells that stained for vWF and DBA-Lectin, suggesting a mixed population which included endothelial-like cells. PBMCs isolated from one pig, that were split and placed in separate wells under different culture conditions, were able to differentiate into ELCs or BOCs. As shown by the immunocytochemistry staining (Figure 2b,d), there is the possibility that ELCs were present in the BOC cultures, however due to the speed BOC emerged and proliferated in culture, it is unlikely that the BOC were present in wells of ELC that presented with a cobblestone-like morphology. Future work would require cell sorting to separate the different populations. Our preliminary mRNA expression data suggest that BOCs under static conditions express CD31 and vWF at approximately 1.5–2% of PAEC levels and over 300-fold higher levels of the mesenchymal marker CD90. These cells may warrant further study in the future in order to confirm their origin and capacity to transdifferentiate into ECs or smooth muscle cells as a possible additional source of cells to be used for vascular graft applications [57,58,59,60].

## 4. Materials and Methods

### 4.1. Blood Collection

Isolation of ELCs was performed using blood collected from White Landrace pigs (55–60 kg weight). Samples were collected at the Translational Biomedical Research Centre (TBRC), Bristol, UK (*n* = 28), in accordance with the Animals (Scientific Procedures) Act 1986, and collection was approved by the University of Bristol ethical review panel under Home Office UK license 70/8975. Blood was collected into 500 mL Teruflex blood collection clinical bags containing citrate phosphate dextrose adenine (CPDA-1, Terumo, Tokyo, Japan) under general anaesthesia (GA), adhering to surgical sterile techniques. We used donor pigs that had not undergone prior procedures (*n* = 16) or control pigs reaching termination point after recruitment in other projects (*n* = 12).

Recruited donor pigs were mostly subjected to a short period of GA (approx. 45 min) before blood harvesting, with the exception of one pig which was subjected to approx. 2 h of GA as it required a magnetic resonance imaging (MRI) scan prior to blood collection. Most of the recruited control pigs (8/12) were exposed to a longer GA period (>1.5 h) as well, as they required MRI scanning prior to blood collection.

In order to ascertain whether GA might have a detrimental effect on ELC isolation, we collected blood samples harvested at room temperature from commercial pigs slaughtered at Langford Abattoir, Bristol, UK, immediately following electrical stunning and severance of the carotid and jugular veins into tubes containing either heparin (1000 IU, Leo Laboratories Ltd., Maidenhead, UK, *n* = 2) or EDTA (5mM, final concentration, Becton Dickinson, Oxford, UK; *n* = 5).

Samples were processed either within 15 min of collection at the TBRC laboratory (*n* = 14) or after 30–45 min (*n* = 21) following transport to our research laboratory.

### 4.2. Cell Isolation

Blood from 19 of the 35 pigs was split into batches for processing under different conditions aimed at refining the isolation protocols, resulting in 57 isolations in total. Time from collection to processing (blood diluted prior to density separation) was noted and classified as immediate (within 15 min) or greater than 30 min.

All cell isolation steps were performed at room temperature. Blood was diluted with an equal volume of DPBS containing 2% fetal bovine serum (FBS; Gibco, Waltham, MA, USA; DPBS_2%_) and 16–17 mL blood suspension layered onto 15mL Ficoll–Paque PLUS 1077 (density 1.077 g/mL; GE-Healthcare, Hatfield, UK) in SepMate 50 tubes (StemCell Technologies, Cambridge, UK). Tubes were centrifuged at 1200× *g* for 10–12 min (*n* = 19) or 20 min (*n* = 38) at room temperature (RT), with the maximum brake applied for deceleration to enrich peripheral blood mononuclear cells (PBMCs) above the Ficoll–Paque layer; 10–12 mL of supernatant plasma (containing platelets) was removed and discarded. The enriched PBMCs were aspirated into another tube, pooled 3:1, washed in DPBS_2%_, and centrifuged at 100–120× *g* for 10 min at room temperature without a brake. The supernatant was discarded and the cell pellets were pooled and washed in DPBS_2%_ and centrifuged at 300× *g* for 8 min (with brake).

### 4.3. Red Blood Cell Lysis

Following PBMC isolation, large amounts of red blood cells remained present in the resulting cell pellets. In 32 of the 57 isolations, RBCs in the pellet were lysed using either RBC Lysing Buffer Hybri-Max (Sigma-Aldrich, Gillingham, UK; *n* = 16) at room temperature for 3 min or Ammonium Chloride Red Blood Cell Lysis Buffer (0.8% NH4Cl, 0.1% KHCO3, 0.004% EDTA, pH 7.2–7.6; *n* = 16) at room temperature for 10 min, mixing gently, before being washed twice in DPBS_2%_ (centrifuged at 500× *g* for 7 min).

### 4.4. Cell Culture Media

Isolated PBMCs were then resuspended in ELC media. Two types of cell culture media were used for cell culture and differentiation into blood outgrowth cells, namely, endothelial growth media-2 (EGM-2, Lonza, Cambridge, UK; *n* = 46) or endothelial cell growth medium-2 (ECGM-2, Promocell, Heidelberg, Germany; *n* = 11). Lonza endothelial cell growth medium-2 Bulletkit (CC-3162; Cambridge, UK) consists of EBM-2 basal medium with the addition of EGM-2 SingleQuots supplements (2% FBS, human epidermal growth factor (hEGF), human fibroblast growth factor (hFGF-B), insulin-like growth factor (R3-IGF-1), vascular endothelial growth factor (VEGF), ascorbic acid, hydrocortisone, heparin, and GA-1000 antibiotics (gentamicin, amphotericin-B)); the concentration of the supplements is proprietary information belonging to Lonza. Additional FBS (heat inactivated (Gibco, Waltham, MA, USA) and passed through 0.22 μm filter) was added for a final concentration of 11.4% for use on cells.

Promocell endothelial cell growth medium-2 kit (Cat# C-22111) consists of endothelial cell basal medium-2 with the addition of all supplements except FBS (epidermal growth factor (recombinant human; 5 ng/mL), basic fibroblast growth factor (recombinant human; 10 ng/mL), insulin-like growth factor (R3-IGF-1; 20 ng/mL), vascular endothelial growth factor-165 (recombinant human; 0.5 ng/mL), ascorbic acid (1 μg/mL), hydrocortisone (0.2 μg/mL), and heparin (22.5 μg/mL). Antibiotics (1% penicillin/streptomycin (P/S), Sigma-Aldrich, Gillingham, UK) were added to supplemented media, and hyclone FBS (GE Healthcare, Hatfield, UK) was added for a final concentration of 9% for use on cells.

Cells were plated onto Type-I rat tail collagen-coated six-well plates (Gibco, Waltham, MA, USA) with cells isolated from approximately 45 mL of blood per well (cells deemed to be Passage 0 and at Day 0).

Cells were allowed to adhere for 24 h, after which time non-adherent cells were removed gently by pipette after gentle swirling of the plate, followed by two washes in DPBS, before 5 mL fresh medium was added. This process was repeated daily until day 5. After this period, medium was changed every 3–5 days. Cells were observed daily to determine the number of days until the appearance of proliferating cells, either with a cobblestone phenotype or otherwise.

After colonies of outgrowth cells were well established, they were passaged by enzymatic detachment with TryplE Express (Gibco, Waltham, MA, USA) at 37 °C and, depending on the size of the colony, transferred to either another collagen-coated six-well plate or expanded into uncoated tissue culture flasks.

### 4.5. ELC Characterisation

#### 4.5.1. Immunocytochemistry

Cells for immunocytochemical staining were seeded at ~60,000 cells/cm^2^ (20,000 cells/well) directly into 96-well plates (flat bottom black polystyrene with µ-clear bottom) and cultured until confluence, where possible, and fixed in 3% paraformaldehyde. The presence of EC markers was assessed with antibodies against platelet EC adhesion molecule (PECAM-1; CD31), vascular endothelial (VE)-cadherin (CD144), and von Willebrand factor (vWF), and stained with Dolichos Biflorus Agglutinin (DBA)-Lectin. Antibodies against smooth muscle myosin heavy chain (SM-MHC) cell and alpha-smooth muscle actin (α-SMA) were used to detect the presence of smooth muscle cell markers, and anti-Thy-1 (CD90) was used to detect mesenchymal or fibroblast markers. Cells were first permeabilised with 0.1% and 0.3% Triton X-100 for anti-vWF and anti-SM-MHC, respectively. An antibody against vimentin was used as well; while this class-III intermediate filament protein has been shown to be present in EC, it is present in other non-epithelial cells as well, especially those of mesenchymal origin such as fibroblasts. CD45 was used as a pan-leucocyte marker to rule out the presence of monocytes in the culture. IgG controls were used during all ICC procedures. Antibody information can be found in Table A3.

#### 4.5.2. Porcine Aortic Endothelial Cell Isolation

For comparative studies, PAECs were isolated from segments of aorta collected from control pigs (*n* = 3) terminated following completion of unrelated in vivo studies at TBRC. Upon collection, aortae were placed into DMEM containing 1% P/S and kept at 4 °C until cell isolation (within 24 h). Aortic rings were cut open longitudinally and ECs were collected by lightly scraping the lumen with a scalpel before transfer into EC media (DMEM incl. GlutaMAX-1, 1 g/L D-Glucose, pyruvate (ThermoFisher Scientific, Waltham, MA, USA), 1% P/S, and 10% FBS). ECs were initially cultured in fibronectin-coated plates (100 µg/mL, Sigma-Aldrich, Gillingham, UK), before being passaged into coated T25 flasks by enzymatic detachment (TryplE Express, Gibco, Waltham, MA, USA), then passaged and expanded in non-coated culture flasks until use in experiments, or cryopreserved at −80 °C for later use (CellBanker 2; Amsbio, Abingdon, UK).

#### 4.5.3. Exposure of ELC to Shear Stress

After cells isolated from pig blood were confirmed to be ELCs by ICC, the effects of fluid flow on blood outgrowth cells compared to PAEC controls (*n* = 3 at P2-P3) was investigated. Cells were seeded into six-well plates at approximately 1500 cells/cm^2^ (1 × 10^5^ cells/well) and allowed to proliferate to confluence. After seeded wells were confluent (at passage five), they were assigned to one of three groups: Static, Low, or High shear stress. Media was removed from all wells and 3 mL of fresh medium was added. Cells were placed onto an orbital shaker (Thermo Scientific 4625 Titer Plate Shaker, with 0.3 cm circular orbit) at low shear stress (both low and high shear stress groups, 2.7 ± 0.3 dyn/cm^2^) at 37 °C, 5% CO_2_ or alongside the shaker under static conditions for 50 ± 7 h. After this time, cells in the Static and Low shear stress groups were removed, washed in DPBS, and lysed in Qiazol (Qiagen, Manchester, UK) before being stored at −80 °C until RNA extraction (described below). Media was replaced for the high shear stress group and returned to the incubator with increased shear stress (10.9 ± 0.2 dyn/cm^2^) for 51 ± 3 h, followed by cell lysis as previously described. See Appendix A for shear stress prediction calculation.

#### 4.5.4. In Vitro Seeding of ELC on Decellularised Human SV (D-hSV)

The seeding potential of ELCs on D-hSV was assessed in vitro. hSVs surplus to requirements at the completion of CABG surgery at the Bristol Heart Institute, UK, were obtained. Tissue was obtained in accordance with ethical approval from the local committee (REC reference number: 10/H0107/63). hSVs were decellularised as previously described [56] in 0.01% (*w*/*v*) Sodium Dodecyl Sulphate on a roller at 60 rpm for 24 h, followed by two washes in DPBS. D-hSVs were then stored in DPBS at 4 °C until use.

Small lengths (approximately 2 cm) of D-hSV (ten lengths from nine patients) were tied at one end with 4–0 silk suture, and 1 × 10^6^ ELCs (*n* = 3 pig sources at Passages 6–9) were carefully pipetted into the lumen in approximately 50 μL ELC media (Lonza, Cambridge, UK). The open end was then tied and the seeded vein placed into a 5 mL polystyrene round-bottomed tube (Corning, Glendale, AZ, USA) with a vented cap in 3 mL ELC media. Tubes were placed onto a roller and elevated to an angle of approximately seven degrees to prevent media spilling from tube at 1 rpm in an incubator (37 °C, 5% CO_2_) for 96 h. After this initial seeding period, the tied ends were cut off and the remainder of the seeded vein returned to the incubator in a six-well plate in 3 mL ELC media under static culture conditions for 72 h.

Following cell seeding and culture, veins were fixed in 10% formalin for at least 16 h at 4 °C before transfer to PBS. Pre-seeding pieces of D-hSV were taken as control.

### 4.6. mRNA Expression

RNA was extracted from cells and purified with the RNeasy mini kit (Qiagen, Manchester, UK) following the manufacturer’s instructions, and RNA yield and purity were assessed using a NanoDrop 1000 spectrophotometer (ThermoScientific, Waltham, MA, USA; Table A1). cDNA was synthesised from 200 ng RNA using either a High-Capacity RNA-to-cDNA™ Kit (Applied Biosystems, Waltham, MA, USA) or a Transcriptor First Strand cDNA Synthesis Kit (Roche, Basel, Switzerland) according to the manufacturer’s instructions. Thermocycling conditions were 37 °C for 60 min, followed by 95 °C for 5 min, then held at 4 °C and stored at −20 °C until use; a 96-well quantitative PCR was then carried out with a LightCycler 480 II (see Table A2 for reaction protocol; Roche, Basel, Switzerland) using a LightCycler 480 SYBR Green 1 Master kit (Roche, Basel, Switzerland). All mRNA levels were normalised to beta-actin (ACTB). ACTB Ct values for all samples were analysed and showed no significant deviation from the mean value using a one-way *t*-test (*p* > 0.05; GraphPad Prism); as such, only one housekeeping gene was deemed necessary. mRNA expression of the generic EC markers PECAM-1 (CD31), VE-Cadherin (CDH5), vWF, and eNOS were determined. mRNA expression was calculated using the *2^–ddCt* method and expression was compared to PAEC at static conditions (Table 3*)* using the dCt calculation (Figure 3). Porcine primer (KiCqStart Predesigned SYBR Green Primers, Sigma-Aldrich, Gillingham, UK) sequences for the aforementioned genes are shown in Table A4, and were used at 1 μM per reaction. No-reverse transcription controls and no-sample (water) controls were included in each qPCR experiment for each gene of interest, and Ct values were always observed to be >40. Due to the low number of cell isolations (*n* = 2–3 per group), statistical comparisons were not performed.

### 4.7. Histology

Tissue was paraffin-embedded and sections of 5 µm thickness were cut. Following heat-mediated antigen retrieval (10 mM Citrate Buffer, pH 6), sections were dual-stained with DBA-lectin and anti-alpha smooth muscle actin-Cy3 (see Table A3 for details) and mounted in Prolong Gold Antifade medium (Life Technologies, Bleiswijk, The Netherlands). Images of the stained vessel cross-sections were opened in ImageJ (v.1.52), the perimeter of the lumen and length of DBA-stained endothelium were measured, and the percentage of lumen coverage was calculated. In order to confirm that ELCs seeded onto D-hSV continued to express CD31, following antigen retrieval, sections were immunostained with anti-CD31 followed by biotinylated goat anti-rabbit IgG (3.5 µg/mL; B7389, Sigma-Aldrich, Gillingham, UK) and detected using a DAB peroxidase system.

## 5. Conclusions

This study shows that it is possible to isolate endothelial-like cells from porcine blood that express markers similar to those seen on PAECs, and that these ELCs can be used during vascular bioengineering to produce an endothelium on acellular vascular conduits. Hence, these cells have great translational potential, as producing a viable endothelium is the first step in reducing early thrombosis and preventing late restenosis after coronary or peripheral artery bypass grafting surgery.

## Figures and Tables

**Figure 1 ijms-23-06633-f001:**
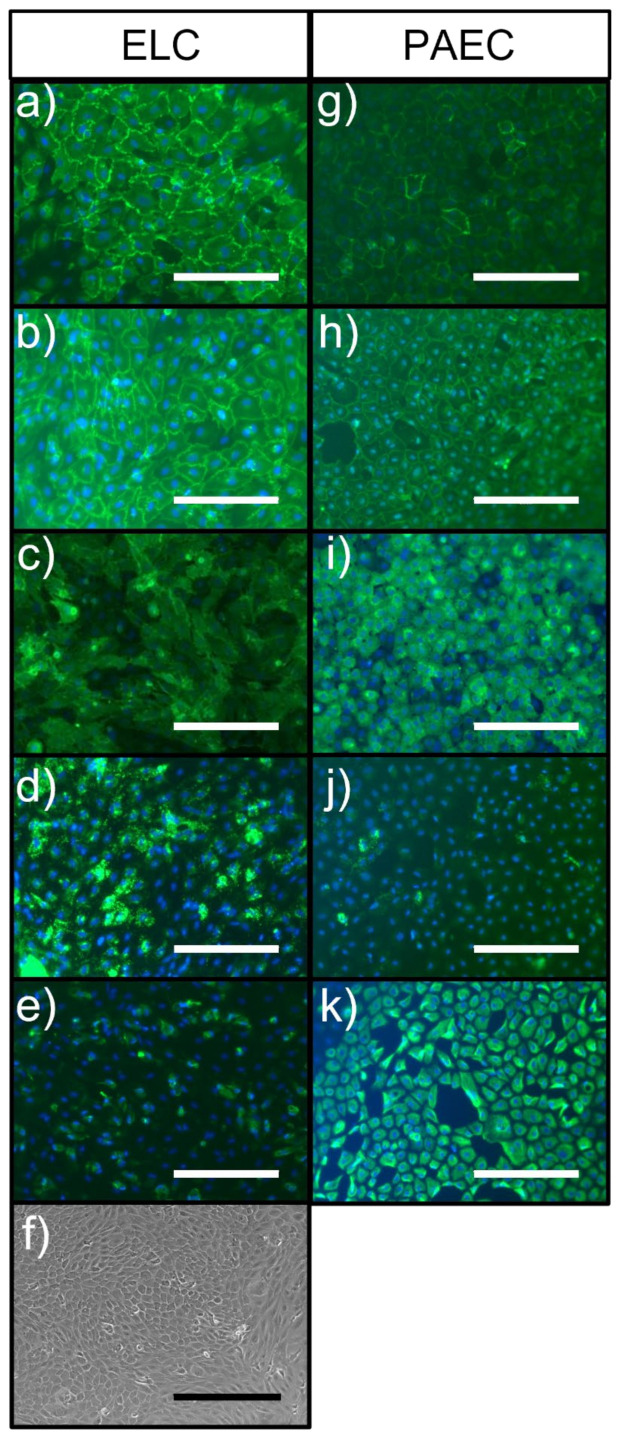
Representative images of porcine endothelial-like cells (ELC; (**a**–**e**)) and porcine aortic endothelial cells (PAEC; (**g**–**k**)) stained for common endothelial cell markers (green): CD31 (**a**,**g**), VE-Cadherin (**b**,**h**), DBA-Lectin (**c**,**i**), von Willebrand factor (vWF, (**d**,**j**)), and vimentin (**e**,**k**); (**f**) phase contrast image of ELC showing cobblestone morphology. Markers of interest are shown in green. Nuclei are stained with DAPI (blue). White bars represent 200 μm. The black bar in (**f**) represents 400 μm.

**Figure 2 ijms-23-06633-f002:**
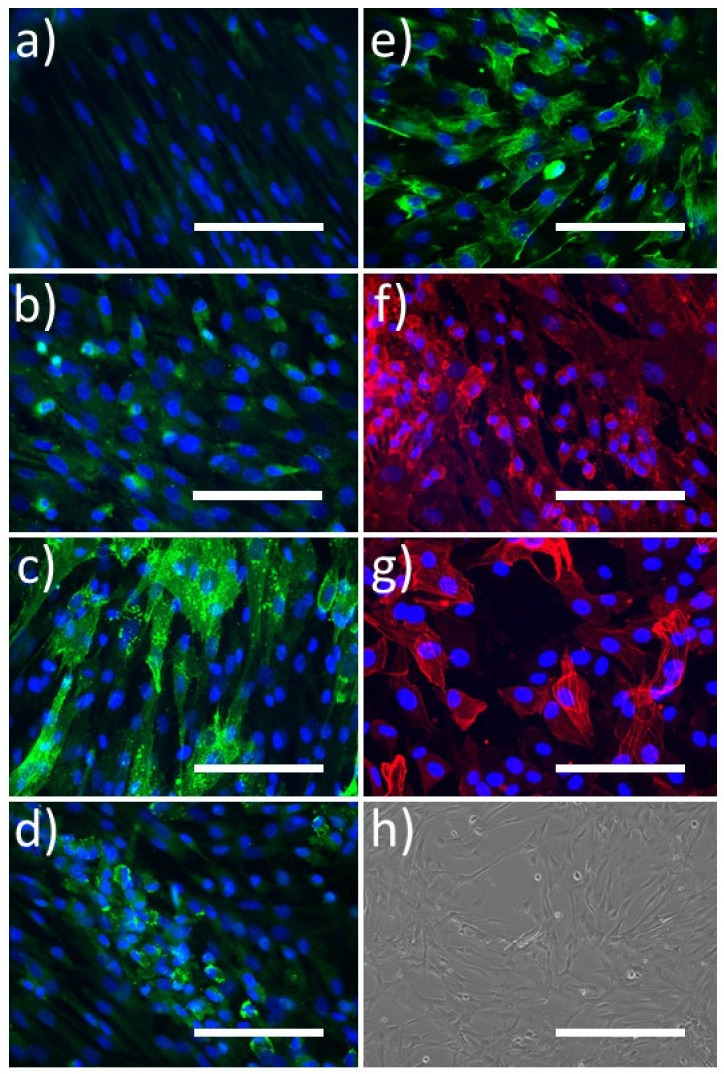
Representative images of porcine blood outgrowth cells (BOC) stained for common endothelial cell markers (green; (**a**–**e**)) and mesenchymal cell markers (red; (**f**,**g**)): (**a**) CD31, (**b**) VE-Cadherin, (**c**) DBA-Lectin, (**d**) von Willebrand factor (vWF), (**e**) vimentin, (**f**) CD90/Thy-1, (**g**) α-smooth muscle actin, (**h**) phase contrast image of blood outgrowth cells showing spindle-like morphology. Nuclei are stained with DAPI (blue). All bars represent 400 μm.

**Figure 3 ijms-23-06633-f003:**
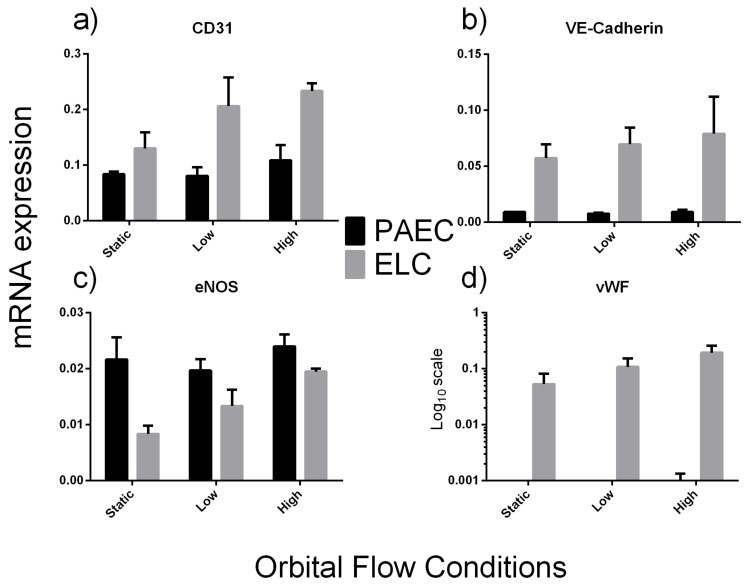
mRNA expression (arbitrary units) of endothelial cell markers CD31 (**a**), VE-Cadherin (**b**), endothelial nitric oxide synthase (eNOS) (**c**) and von willebrand factor (vWF) (**d**) of porcine endothelial-like cells (ELC) and aortic endothelial cells (PAEC) under different flow conditions on the orbital shaker. Error bars are standard errors of the mean.

**Figure 4 ijms-23-06633-f004:**
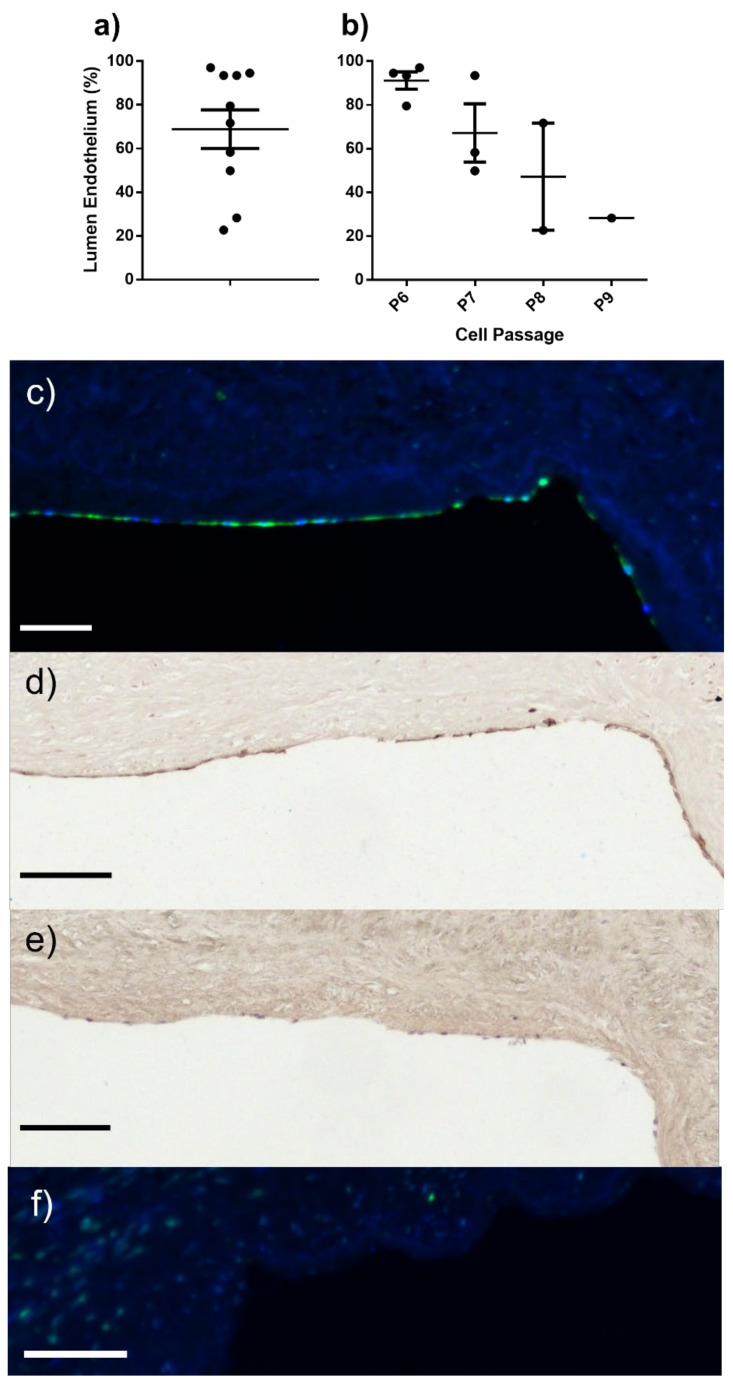
Seeding endothelial-like cells (ELC) onto decellularised human saphenous vein. (**a**) Percentage of lumen covered by ELC following 96 h on a roller at 1 rpm and a further 72 h in static culture in (**a**) all seeded vessels and (**b**) lumen coverage for each cell passage used. Error bars indicate standard error of the mean. Representative serial section images of ELC on lumen, stained with (**c**,**f**) DBA-Lectin (green) and DAPI (blue), (**d**) CD31 (dark brown), and (**e**) H&E for nuclei. (**f**) Unseeded decellularised human saphenous vein. Black and white scale bars represent 100 μm.

**Table 1 ijms-23-06633-t001:** Conditions used for isolating ELC from blood taken under surgical conditions.

Isolations	Previous Procedure	Anaesthetic	Transport	Time to Isolation	Centrifuge Time	RBC Lysis	Media	Cell Growth		
								ELC	BOC	No Growth
28 Pigs = 43 Batches	Yes 12 Pigs 17 Batches	Short 4 Pigs 4 Batches	Yes 4 Pigs 4 Batches	>30 min 4 Pigs 4 Batches	10–12 min 4 Pigs 4 Batches	Yes 3 Pigs 3 Batches	Lonza 3 Pigs 3 Batches	1 *	0	2
						No 1 Pig 1 Batch	Lonza 1 Pig 1 Batch	0	0	1
		Long 8 Pigs 13 Batches	Yes 6 Pigs 10 Batches	>30 min 6 Pigs 10 Batches	10–12 min 6 Pigs 10 Batches	Yes 6 Pigs 9 Batches	Lonza 2 Pigs 3 Batches	0	0	3
							Promocell 4 Pigs 6 Batches	0	0	6
						No 1 Pig 1 Batch	Promocell 1 Pig 1 Batch	0	0	1
			No 2 Pigs 3 Batches	Immediate 1 Pig 1 Batch	20 min 1 Pig 1 Batch	No 1 Pig 1 Batch	Lonza 1 Pig, 1 Batch	0	0	1
				>30 min 1 Pig 2 Batches	20 min 1 Pig, 2 Batches	Yes 1 Pig, 1 Batch	Lonza 1 Pig 1 Batch	0	0	1
						No 1 Pig 1 Batch	Lonza 1 Pig 1 Batch	0	0	1
	No 16 Pigs 26 Batches	Short 15 Pigs 24 Batches	Yes 4 Pigs 6 Batches	>30 min 4 Pigs 6 Batches	10–12 min 3 Pigs 5 Batches	Yes 3 Pigs 4 Batches	Lonza 1 Pig 1 Batch	0	1	0
							Promocell 2 Pigs 3 Batches	0	0	3
						No 1 Pig 1 Batch	Promocell 1 Pig 1 Batch	0	0	1
					20 min 1 Pig 1 Batch	Yes 1 Pig 1 Batch	Lonza 1 Pig 1 Batch	0	0	1
			No 11 Pigs 18 Batches	Immediate 7 Pigs 9 Batches	20 min 7 Pigs 9 Batches	Yes 7 Pigs 7 Batches	Lonza 7 Pigs 7 Batches	4	1	2
						No 2 Pigs 2 Batches	Lonza 2 Pigs 2 Batches	2	0	0
				>30 min 5 Pigs 9 Batches	20 min 5 Pigs 9 Batches	Yes 5 pigs 6 Batches	Lonza 5 pigs 6 Batches	1 *	4	1
						No 3 Pigs 3 Batches	Lonza 3 pigs 3 Batches	0	2	1
		Long 1 Pig 2 Batches	No 1 Pig 2 Batches	>30 min 1 Pig 2 Batches	20 min 1 Pig 2 Batches	Yes 1 Pig 1 Batch	Lonza 1 Pig 1 Batch	1 *	0	0
						No 1 Pig 1 Batch	Lonza 1 Pig 1 Batch	1 *	0	0

* indicates that while cobblestone-morphology ELCs were derived, they did not proliferate well in culture, and colonies could not be scaled up for use. Red blood cell (RBC), endothelial-like cell (ELC), blood outgrowth cell (BOC).

**Table 2 ijms-23-06633-t002:** Conditions used for isolating blood outgrowth cells from blood collected from pigs at commercial abattoir.

Isolations	Anticoagulant	Previous Procedure	Anaesthetic	Transport	Time to Isolation	Centrifuge Time	RBC Lysis	Media	Cell Growth		
									ELC	BOC	No Growth
5 pigs 10 Batches	EDTA	No	None	Yes	>30 min	20 min	No	Lonza	0	10	0
2 Pigs 4 Batches	Heparin	No	None	Yes	>30 min	20 min	No	Lonza	0	4	0

Red blood cell (RBC), endothelial-like cell (ELC), blood outgrowth cell (BOC), ethylenediaminetetraacetic acid (EDTA).

**Table 3 ijms-23-06633-t003:** Fold-change in mRNA expression of classical endothelial cell (EC) markers under static, low, or high flow conditions. Expression shown as fold change from aortic ECs under static conditions. Mean ± SEM.

		CD31	VE-Cadherin	eNOS	vWF
Aortic Endothelial Cells (PAEC)	Static (*n* = 3)	1.0 ± 0.0	1.0 ± 0.0	1.0 ± 0.2	1.3 ± 0.6
	Low (*n* = 3)	1.0 ± 0.2	0.9 ± 0.1	0.9 ± 0.1	0.5 ± 0.2
	High (*n* = 3)	1.3 ± 0.3	1.0 ± 0.2	1.2 ± 0.1	3.6 ± 3.0
Endothelial-like Cells (ELC)	Static (*n* = 3)	1.6 ± 0.3	6.5 ± 1.4	0.4 ± 0.1	300.6 ± 155.1
	Low (*n* = 3)	2.5 ± 0.6	7.8 ± 1.7	0.7 ± 0.1	609.6 ± 251.0
	High (*n* = 2)	2.8 ± 0.2	8.8 ± 3.7	0.9 ± 0.0	1088.1 ± 356.3

Vascular endothelial (VE)-Cadherin, endothelial nitric oxide synthase (eNOS), von willebrand factor (vWF).

## Data Availability

All datasets presented in this manuscript can be made available upon reasonable request.

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
