# Peer review of "Development and Preliminary Testing of Porcine Blood-Derived Endothelial-like Cells for Vascular Tissue Engineering Applications: Protocol Optimisation and Seeding of Decellularised Human Saphenous Veins"

_ijms, 2022, doi:10.3390/ijms23126633_

Round 1
Reviewer 1 Report
The manuscript entitled “Development and preliminary testing of porcine endothelial colony forming cells for vascular tissue engineering applications: protocol optimization and seeding of decellularized human saphenous veins” is devoted to the important problem of vascular tissue engineering. The manuscript is interesting and well-written and designed but there are some minor methodological issues that must be solved before publication.
- RNA extraction, cDNA synthesis and gene expression analysis must be described in the separate paragraph of Material and Methods section not in 4.5.3. Exposure of ECFC to shear stress;
- Quantitative real-time PCR results must be presented according to MIQE Guidelines (The MIQE Guidelines: Minimum Information for publication of Quantitative Real-Time PCR Experiments, Clinical Chemistry 55:4, 611-622, 2009);
- Did authors access the integrity (RIN index or analogues) and quality of RNA? This information is obligatory for gene expression experiments;
- What method was used for gene expression calculation (2^–ddCq, Pfaffl, etc.). It is very important for understanding presenting results and it is necessary that other authors be able to compare their results with the results obtained in this article;
- The authors must normalize the PCR results when assessing gene expression not for 1, but for 3-5 reference genes. The expression of reference genes varies quite a lot and this can have a strong influence on the genes of interest expression. In order to avoid incorrect results, it is necessary to normalize the PCR results to the geometric mean of several references (J. Vandesompele, K. De Preter, F. Pattyn, B. Poppe, N.V. Roy, A. De Paepe, F. Speleman, Accurate normalization of real-time quantitative RT-PCR data by geometric averaging of multiple internal control genes, Genome Biol. 3 (2002) research0034.1);
- I suggest using not fold-change but arbitrary units calculated by the dCq Method (use free BioRad Real-Time PCR Application Guide) for gene expression results in Figure 3. It is more correct according to the statistical analysis, and the fold-change results are already presented in the Table 3;
- Statistical analysis methods must be presented in the Material and Methods section.
Author Response
Reviewer #1:
1. RNA extraction, cDNA synthesis and gene expression analysis must be described in the separate paragraph of Material and Methods section not in 4.5.3. Exposure of ECFC to shear stress; We thank the reviewer for this suggestion; a separate ‘mRNA expression’ paragraph has now been included starting at Line 478.
2. Quantitative real-time PCR results must be presented according to MIQE Guidelines (The MIQE Guidelines: Minimum Information for publication of Quantitative Real-Time PCR Experiments, Clinical Chemistry 55:4, 611-622, 2009); thanks - we have now consulted the MIQE guidelines, and where possible applied the guidance as shown within the paragraphs starting at Line 478 (especially Lines 484, 486, 488, 492 and 496) and included new tables (Table A1 & A2 – Lines 545 and 548, respectively) with the required information. However, we would like to highlight that in our study the inclusion of mRNA expression was utilised to confirm expression of endothelial markers, rather than there being a requirement for precise quantification.
3. Did authors access the integrity (RIN index or analogues) and quality of RNA? This information is obligatory for gene expression experiments; The RNA quality and yield were assessed for 260/280 and 260/230 ratios respectively using a NanoDrop 1000, and this information has been added to Table A1 for each sample (line 545), and the text modified accordingly at Line 481.
4. What method was used for gene expression calculation (2^–ddCq, Pfaffl, etc.). It is very important for understanding presenting results and it is necessary that other authors be able to compare their results with the results obtained in this article; We have followed the reviewers guidance, and the BioRad Real-Time PCR Application Guide, and presented data calculated using 2^–ddCt in Table 3 (Line 151), and in Figure 3 (Line 144) the data is expressed using the dCt method. Relevant text has been updated at Lines 143 and 493.
5. The authors must normalize the PCR results when assessing gene expression not for 1, but for 3-5 reference genes. The expression of reference genes varies quite a lot and this can have a strong influence on the genes of interest expression. In order to avoid incorrect results, it is necessary to normalize the PCR results to the geometric mean of several references (J. Vandesompele, K. De Preter, F. Pattyn, B. Poppe, N.V. Roy, A. De Paepe, F. Speleman, Accurate normalization of real-time quantitative RT-PCR data by geometric averaging of multiple internal control genes, Genome Biol. 3 (2002) research0034.1); We are thankful appreciate the necessity for multiple reference genes. However, we have assessed all our samples for variation in the reference gene (ACTB) to determine the validity of the use of this particular reference gene in our cell types. The ACTB Ct expression did not significantly deviate from the mean values indicating consistency of expression of this reference gene, and therefore we consider it justified to use just one reference gene in this study. We have outlined the lack of variation in the reference gene at Line 488. Moreover, as mentioned above, the addition of mRNA expression was seen as a way of confirming expression of endothelial markers, rather than an accurate quantification.
6. I suggest using not fold-change but arbitrary units calculated by the dCq Method (use free BioRad Real-Time PCR Application Guide) for gene expression results in Figure 3. It is more correct according to the statistical analysis, and the fold-change results are already presented in the Table 3; Thanks - we have modified Figure 3 (Line 145) and accompanying figure legend (line 146), to show the results using the dCq method.
7. Statistical analysis methods must be presented in the Material and Methods section. P-values were included in the previous draft in error and have now been removed (Paragraph at Line 149). Due to the low numbers assessed (n=2-3), we were unable to perform meaningful statistical analysis, hence no statistical analysis methods are included. A statement to this effect is already in Line 498, from our original submission.
Reviewer 2 Report
The authors propose a protocol to use ECFCs extracted from pig peripheral blood for therapeutic purposes. To demonstrate their therapeutic potential, they propose a model in which they cover a decellularized mammary artery with this ECFCs obtained from pig blood.
The approach is interesting, but the methodology and the characterization of the ECFCs seem confusing and limited to me. Before continuing to raise other technical aspects, it seems essential to me to answer the following questions.
- The authors note that they only obtain ECFCs from 20% of their donor pigs but acknowledge that they have several types of donor pigs, where they have followed different approaches. This approach is confusing, the variety in its results could be due to both the approaches used and the conditions of the pigs used. I believe that the authors should simplify this proof of concept and choose a single type of donor, with which they can justify the best strategy to follow to obtain functional ECFCs.
- The characterization of the purified ECFCs seems insufficient to me. By definition, an Endothelial colony-forming cells (ECFCs) is characterized by its clonogenic ability (DOI: 10.1182/blood-2004-04-1396), large expansion potential, stable endothelial phenotype, and robust vessel-forming capacity (DOI: 10.1182/blood-2006-08-043471 / DOI: 10.1002/9780470151808.sc02c01s6). With a simple analysis of surface markers such as (CD31, VE-Cadherin, von Willebrand factor, etz), the authors cannot distinguish between mature endothelial cells, circulating endothelial cells, endothelial progenitor cells also known as angiogenic cells, or ECFCs. The authors should carry out additional tests such as test of clonal ability. These tests would also allow them to know what type of ECFC they have and even separate the cells with the greatest clonal ability, allowing them to obtain more homogeneous populations.
- Authors should better review the literature. Since David A Ingram first characterized ECFCs in 2004 (DOI: 10.1182/blood-2004-04-1396), more than 500 papers have been published with the nomenclature ECFCs (Title/Abstract). Only in the last 5 years, more than 250 have been written.
Sometimes I am not sure if the authors know how to distinguish between EPC, CAC, ECFC, or other nomenclatures, associated with the putative ECFCs. The case that has caught my attention the most is the one in this quote [28] (line 173: Estimates calculate that there are 2.6 ± 1.6 parent ECFCs/ml [28]). The authors base their results on the approximation that the NEJM makes of the number of ECFCs in the blood, but in reality, in this work they are talking about circulating endothelial cells, probably detached from the walls of the arteries, mature or senescent endothelial cells, but not ECFCs. Other examples: [13], [16], [30].
- When ECFCs are purified from blood or other tissues, they must be adequately separated from the rest of the cell types, for their subsequent expansion and characterization. I am concerned about some comments from the authors, in which they claim to have found other populations of cells in the culture dishes with higher proliferative rate.
Line 120: “Cells that did not have a cobblestone morphology (blood outgrowth cells (BOC), grew at a higher proliferative rate, tended to grow in multilayers and were positive for CD90, vimentin and alpha-SMC, contained small patches of cells expressing vWF, DBAlectin, and VE-Cadherin, but were negative for CD31, SM-MHC and CD45”.
Could the authors tell me if they have adequately separated the ECFCs from the rest of the identified cells, before their expansion and characterization? Their co-culture could compromise the clonal expansion and functional properties of the purified ECFCs.
- Line 92: “PBMCs from 1 pig went onto differentiate into both ECFCs and BOCs, depending on the culture conditions.” This statement should have adequate validation. The authors have not shown that PBMCs convert to ECFCs. Only have validated the presence of cells with an endothelial phenotype after isolating cells by density gradient.
- Line 250: “vWF staining was higher in the ECFCs than PAECs, and mRNA expression confirmed this finding. There was a large amount of variability between mRNA expression in ECFC, as highlighted by the large SEMs. This is in keeping with previous studies and is thought to be a reflection of maximum cell density [47]”.
I believe that this is associated with the heterogeneity of the different populations of ECFCs that are purified from blood. To avoid this heterogeneity, the authors could separate the populations by groups with the same clonal ability and would have more homogeneous cell populations.
- Line 26: “we confirmed the feasibility of seeding derived porcine ECFC onto a decellularised human vein scaffold with approximately 90% lumen coverage at lower passages, and show that increasing cell passage results in reduced endothelial coverage”. The authors must show images that allow us to visualize the new endothelial cells lining the mammary artery.
Author Response
We thank the reviewer for their constructive and useful comments on this manuscript. We have revised our manuscript taking these into consideration, and the responses to each point are below including the details of any changes made to the manuscript text. The manuscript sections referred to in the responses below have also been highlighted in the corrected manuscript file attached.
Reviewer #2:
- The authors note that they only obtain ECFCs from 20% of their donor pigs but acknowledge that they have several types of donor pigs, where they have followed different approaches. This approach is confusing, the variety in its results could be due to both the approaches used and the conditions of the pigs used. I believe that the authors should simplify this proof of concept and choose a single type of donor, with which they can justify the best strategy to follow to obtain functional ECFCs.
We thank the reviewer for their comment, and shall try to justify our approach here.
To prevent unnecessary use of additional pigs and to act in line with the NC3Rs principles on animal welfare, blood samples were collected at termination from control cases of multiple studies being undertaken in the Translational Biomedical Research Centre. However, in all these cases the same breed of pig is used, with similar weight and age range. During the study period, there was a global shortage of EGM media, and the global COVID pandemic, and as such, protocols were forced to be modified and adapted, hence the different approaches. A paragraph has been added at Line 171 to better highlight this.
As we only obtained endothelial-like cells from a few of these samples, we consider it best to include the different protocols, as our study and the different approaches used suggest that very specific conditions are necessary to isolate these cells; we reflected that their inclusion is of interest and benefit to researchers and future implementation of this methodology.
In addition, blood was also taken from pigs at the point of slaughter in a commercial abattoir as, if this approach was feasible, it would reduce the number of animals needed and circumvent the requirement for porcine “clinical grade” blood at termination of studies, which would be a more accessible supply of blood, and would also reduce costs. Accordingly, we have ruled out several different approaches, and therefore our findings will benefit the reproducibility of future studies. Text has been added from Line 262 to reflect this.
- The characterization of the purified ECFCs seems insufficient to me. By definition, an Endothelial colony-forming cells (ECFCs) is characterized by its clonogenic ability (DOI: 10.1182/blood-2004-04-1396), large expansion potential, stable endothelial phenotype, and robust vessel-forming capacity (DOI: 10.1182/blood-2006-08-043471 / DOI: 10.1002/9780470151808.sc02c01s6). With a simple analysis of surface markers such as (CD31, VE-Cadherin, von Willebrand factor, etz), the authors cannot distinguish between mature endothelial cells, circulating endothelial cells, endothelial progenitor cells also known as angiogenic cells, or ECFCs. The authors should carry out additional tests such as test of clonal ability. These tests would also allow them to know what type of ECFC they have and even separate the cells with the greatest clonal ability, allowing them to obtain more homogeneous populations.
We thank the reviewer for their insightful comments and apologise for the confusion in cell-type nomenclature. As our ultimate aim is to produce endothelial-like cells from the blood that have potential for tissue engineering translational studies, we feel the nomenclature isn't as important as their practical application. The markers we used are widely reported to be deployed for identifying endothelial cells, and as our cell type showed similar expression to “native” ECs isolated from healthy porcine arteries, we have altered the text, and now call them the more generic “endothelial-like cells” (ELCs) throughout (all instances highlighted in the manuscript text).
With the resources available we were not able to test for clonal ability, or to differentiate between mature endothelial cells, circulating endothelial cells, endothelial progenitor cells, or ECFCs, but this would be a good way forward in future studies. We feel the ability of the cells to maintain these markers for multiple passages suggests they are appropriate for use in translational tissue engineering studies. As such, we have modified the text in the discussion to highlight this starting at Line 176.
- Authors should better review the literature. Since David A Ingram first characterized ECFCs in 2004 (DOI: 10.1182/blood-2004-04-1396), more than 500 papers have been published with the nomenclature ECFCs (Title/Abstract). Only in the last 5 years, more than 250 have been written.
Sometimes I am not sure if the authors know how to distinguish between EPC, CAC, ECFC, or other nomenclatures, associated with the putative ECFCs. The case that has caught my attention the most is the one in this quote [28] (line 173: Estimates calculate that there are 2.6 ± 1.6 parent ECFCs/ml [28]). The authors base their results on the approximation that the NEJM makes of the number of ECFCs in the blood, but in reality, in this work they are talking about circulating endothelial cells, probably detached from the walls of the arteries, mature or senescent endothelial cells, but not ECFCs. Other examples: [13], [16], [30].
We thank the reviewer for their helpful suggestion and having revisited the published literature now realise that we were inconsistent with our use of nomenclature for putative ECFCs. Where necessary we have now modified the text to more accurately to portray the different cell types discussed (Line 191 onwards). As we are not 100% sure of the outgrowth endothelial cell type, we have isolated, due to a lack of assessing clonal ability, we have modified the text to replace ECFC with endothelial-like cell (ELC) throughout (including Figures 1 and 3 at lines 128 and 145 respectively), and have also modified the manuscript title to reflect this. The new proposed title is ‘Development and preliminary testing of porcine blood derived endothelial-like cells for vascular tissue engineering applications: protocol optimisation and seeding of decellularised human saphenous veins’.
- When ECFCs are purified from blood or other tissues, they must be adequately separated from the rest of the cell types, for their subsequent expansion and characterization. I am concerned about some comments from the authors, in which they claim to have found other populations of cells in the culture dishes with higher proliferative rate.
Line 120: “Cells that did not have a cobblestone morphology (blood outgrowth cells (BOC), grew at a higher proliferative rate, tended to grow in multilayers and were positive for CD90, vimentin and alpha-SMC, contained small patches of cells expressing vWF, DBAlectin, and VE-Cadherin, but were negative for CD31, SM-MHC and CD45”.
Could the authors tell me if they have adequately separated the ECFCs from the rest of the identified cells, before their expansion and characterization? Their co-culture could compromise the clonal expansion and functional properties of the purified ECFCs.
We thank the reviewer for their comments. Our method of endothelial-like cell expansion relied on the initial isolation of PBMCs by density gradient separation, subsequent culture of adhered monocytes, and the emergence of colony forming cells that were subsequently cultured further. Non-cobblestone like cells appeared in separate wells and, where present, rapidly proliferated. Cells which formed colonies appeared in separate wells at much later time points, and were expanded, and resulted in a cobblestone like morphology.
Due to the speed the BOC grew, and how early they were detected, it was highly unlikely they would have been present in wells where we observed endothelial-like cells with a cobblestone morphology. There was a potential for ELC presence in BOC wells, as already discussed in the text, but these were not used in this study. We believe that due to the potential these BOC cells may have in future studies, the images and discussion should be included. We have modified the text to highlight this (Line 324 onwards).
- Line 92: “PBMCs from 1 pig went onto differentiate into both ECFCs and BOCs, depending on the culture conditions.” This statement should have adequate validation. The authors have not shown that PBMCs convert to ECFCs. Only have validated the presence of cells with an endothelial phenotype after isolating cells by density gradient.
Thank you for highlighting this, and apologies for the confusion. The PBMCs were in separate wells and one well differentiated into what we are now calling endothelial-like cells, and the other into blood outgrowth cells. The text in the results section has been updated to reflect this (Line 96).
- Line 250: “vWF staining was higher in the ECFCs than PAECs, and mRNA expression confirmed this finding. There was a large amount of variability between mRNA expression in ECFC, as highlighted by the large SEMs. This is in keeping with previous studies and is thought to be a reflection of maximum cell density [47]”.
I believe that this is associated with the heterogeneity of the different populations of ECFCs that are purified from blood. To avoid this heterogeneity, the authors could separate the populations by groups with the same clonal ability and would have more homogeneous cell populations.
We thank you for noting this valid point, and this is definitely something that could be investigated further in future studies. The manuscript text has been modified highlighting this (Line 279-282).
- Line 26: “we confirmed the feasibility of seeding derived porcine ECFC onto a decellularised human vein scaffold with approximately 90% lumen coverage at lower passages, and show that increasing cell passage results in reduced endothelial coverage”. The authors must show images that allow us to visualize the new endothelial cells lining the mammary artery.
We agree this suggestion will add weight to the manuscript. Figure 4 (Line 162) and it’s corresponding legend (Line 163) have now been updated to also include images of serial sections of seeded human saphenous vein, stained with DBA-lectin for endothelial-like cells lining the lumen, CD31, and H&E for cell nuclei. The methods have been updated to include CD31 immunohistochemistry (Line 507-510), and a line of text added to results to reflect this (Line 157).
Round 2
Reviewer 2 Report
Dear authors,
My comments and suggestions can be found after each question and answer made in the first revision of this work .
Best regards
Author response to report 1:
Author's Notes
We thank the reviewer for their constructive and useful comments on this manuscript. We have revised our manuscript taking these into consideration, and the responses to each point are below including the details of any changes made to the manuscript text. The manuscript sections referred to in the responses below have also been highlighted in the corrected manuscript file attached.
Reviewer #2:
- The authors note that they only obtain ECFCs from 20% of their donor pigs but acknowledge that they have several types of donor pigs, where they have followed different approaches. This approach is confusing, the variety in its results could be due to both the approaches used and the conditions of the pigs used. I believe that the authors should simplify this proof of concept and choose a single type of donor, with which they can justify the best strategy to follow to obtain functional ECFCs.
We thank the reviewer for their comment, and shall try to justify our approach here.
To prevent unnecessary use of additional pigs and to act in line with the NC3Rs principles on animal welfare, blood samples were collected at termination from control cases of multiple studies being undertaken in the Translational Biomedical Research Centre. However, in all these cases the same breed of pig is used, with similar weight and age range. During the study period, there was a global shortage of EGM media, and the global COVID pandemic, and as such, protocols were forced to be modified and adapted, hence the different approaches. A paragraph has been added at Line 171 to better highlight this.
As we only obtained endothelial-like cells from a few of these samples, we consider it best to include the different protocols, as our study and the different approaches used suggest that very specific conditions are necessary to isolate these cells; we reflected that their inclusion is of interest and benefit to researchers and future implementation of this methodology.
In addition, blood was also taken from pigs at the point of slaughter in a commercial abattoir as, if this approach was feasible, it would reduce the number of animals needed and circumvent the requirement for porcine “clinical grade” blood at termination of studies, which would be a more accessible supply of blood, and would also reduce costs. Accordingly, we have ruled out several different approaches, and therefore our findings will benefit the reproducibility of future studies. Text has been added from Line 262 to reflect this.
Reviewer: I understand the difficulties that the pandemic has caused you. However, I still think that the different approaches followed may have compromised the results. This would imply that their conclusions could be wrong. I think that the authors should simplify the confounding variables.
The characterization of the purified ECFCs seems insufficient to me. By definition, an Endothelial colony-forming cells (ECFCs) is characterized by its clonogenic ability (DOI: 10.1182/blood-2004-04-1396), large expansion potential, stable endothelial phenotype, and robust vessel-forming capacity (DOI: 10.1182/blood-2006-08-043471 / DOI: 10.1002/9780470151808.sc02c01s6). With a simple analysis of surface markers such as (CD31, VE-Cadherin, von Willebrand factor, etz), the authors cannot distinguish between mature endothelial cells, circulating endothelial cells, endothelial progenitor cells also known as angiogenic cells, or ECFCs. The authors should carry out additional tests such as test of clonal ability. These tests would also allow them to know what type of ECFC they have and even separate the cells with the greatest clonal ability, allowing them to obtain more homogeneous populations.
We thank the reviewer for their insightful comments and apologise for the confusion in cell-type nomenclature. As our ultimate aim is to produce endothelial-like cells from the blood that have potential for tissue engineering translational studies, we feel the nomenclature isn't as important as their practical application. The markers we used are widely reported to be deployed for identifying endothelial cells, and as our cell type showed similar expression to “native” ECs isolated from healthy porcine arteries, we have altered the text, and now call them the more generic “endothelial-like cells” (ELCs) throughout (all instances highlighted in the manuscript text).
Reviewer: Proper identification (nomenclature) is essential, among other reasons, so that other authors can subsequently reproduce the results observed in this study.
The lack of proper identification has been one of the main problems in the study of endothelial cells and their potential in recent years. Different authors have spent years demanding an adequate characterization of these cells with therapeutic potential (Stem Cells Transl Med. 2017 May; 6(5):1316-1320. doi: 10.1002/sctm.16-0360).
Any endothelial cell (progenitor or mature) and even other cells with phenotypes similar to those of endothelial cells (EPC, CAC,...), have the ability to adhere to different substrates, including the walls of arteries and veins. In this sense, in addition to a correct identification (which is not limited to surface markers), it is necessary to carry out additional tests that demonstrate that these cells are putative endothelial cells and that after purification and expansion it’s remain functional. There are many tests that validate the functionality of endothelial cells, some of the most used are tests such as Proliferation; Migration in response to VEGF-A and FGF-2 ; Angiogenesis assays (representative micrograph of endothelial cells-lined capillary-like network on a Matrigel coated plate); Responsiveness to inflammatory stimuli (Up-regulation of E-selectin and ICAM-1 in response to TNF-a); etc.
However, I think this work would make sense if the authors could show that they have purified ECFCs. If the authors have ECFC, they should additionally prove it with clonal ability tests. If not, they should perform additional tests, to show that they are not using other endothelial-like cells or even mature endothelial cells whose expansion could be compromising their functionality.
By definition, an ECFC is characterized by its clonogenic ability, large expansion potential, stable endothelial phenotype, and robust vessel-forming capacity. These are in concordance with the set of cell criteria that, ideally, should follow future vascular cell therapies and tissue engineering applications: (i) easy to isolate and purify; (ii) large proliferative potential; and (iii) robust vessel-forming ability [Tissue Eng Part A. 2009;15:3473-3486]
With the resources available we were not able to test for clonal ability, or to differentiate between mature endothelial cells, circulating endothelial cells, endothelial progenitor cells, or ECFCs, but this would be a good way forward in future studies. We feel the ability of the cells to maintain these markers for multiple passages suggests they are appropriate for use in translational tissue engineering studies. As such, we have modified the text in the discussion to highlight this starting at Line 176.
Reviewer: If the purified cells are ECFC, clonal ability tests are very simple and inexpensive. They only require the medium that was used for their purification and expansion. Colony formation is very easy to quantify. There are different protocols in the literature.
Authors should better review the literature. Since David A Ingram first characterized ECFCs in 2004 (DOI: 10.1182/blood-2004-04-1396), more than 500 papers have been published with the nomenclature ECFCs (Title/Abstract). Only in the last 5 years, more than 250 have been written.
Sometimes I am not sure if the authors know how to distinguish between EPC, CAC, ECFC, or other nomenclatures, associated with the putative ECFCs. The case that has caught my attention the most is the one in this quote [28] (line 173: Estimates calculate that there are 2.6 ± 1.6 parent ECFCs/ml [28]). The authors base their results on the approximation that the NEJM makes of the number of ECFCs in the blood, but in reality, in this work they are talking about circulating endothelial cells, probably detached from the walls of the arteries, mature or senescent endothelial cells, but not ECFCs. Other examples: [13], [16], [30].
We thank the reviewer for their helpful suggestion and having revisited the published literature now realise that we were inconsistent with our use of nomenclature for putative ECFCs. Where necessary we have now modified the text to more accurately to portray the different cell types discussed (Line 191 onwards). As we are not 100% sure of the outgrowth endothelial cell type, we have isolated, due to a lack of assessing clonal ability, we have modified the text to replace ECFC with endothelial-like cell (ELC) throughout (including Figures 1 and 3 at lines 128 and 145 respectively), and have also modified the manuscript title to reflect this. The new proposed title is ‘Development and preliminary testing of porcine blood derived endothelial-like cells for vascular tissue engineering applications: protocol optimisation and seeding of decellularised human saphenous veins’.
Reviewer: As I have commented in the previous section, the authors must better define what type of cells they are using. Many types of cells identified as endothelial cells have been identified in the blood. Among other reasons, the great variety of nomenclatures often makes reproducibility unfeasible.
- When ECFCs are purified from blood or other tissues, they must be adequately separated from the rest of the cell types, for their subsequent expansion and characterization. I am concerned about some comments from the authors, in which they claim to have found other populations of cells in the culture dishes with higher proliferative rate.
Line 120: “Cells that did not have a cobblestone morphology (blood outgrowth cells (BOC), grew at a higher proliferative rate, tended to grow in multilayers and were positive for CD90, vimentin and alpha-SMC, contained small patches of cells expressing vWF, DBAlectin, and VE-Cadherin, but were negative for CD31, SM-MHC and CD45”.
Could the authors tell me if they have adequately separated the ECFCs from the rest of the identified cells, before their expansion and characterization? Their co-culture could compromise the clonal expansion and functional properties of the purified ECFCs.
We thank the reviewer for their comments. Our method of endothelial-like cell expansion relied on the initial isolation of PBMCs by density gradient separation, subsequent culture of adhered monocytes, and the emergence of colony forming cells that were subsequently cultured further. Non-cobblestone like cells appeared in separate wells and, where present, rapidly proliferated. Cells which formed colonies appeared in separate wells at much later time points, and were expanded, and resulted in a cobblestone like morphology.
Due to the speed the BOC grew, and how early they were detected, it was highly unlikely they would have been present in wells where we observed endothelial-like cells with a cobblestone morphology. There was a potential for ELC presence in BOC wells, as already discussed in the text, but these were not used in this study. We believe that due to the potential these BOC cells may have in future studies, the images and discussion should be included. We have modified the text to highlight this (Line 324 onwards).
Reviewer: I don't understand this answer.
If the authors seeded the PBMCs in culture plates (without additional separations), how could different populations appear in different wells? In my experience when I have purified EPC or ECFC from PBMCs, both populations appear in the same culture dish, but at different times.
- Line 92: “PBMCs from 1 pig went onto differentiate into both ECFCs and BOCs, depending on the culture conditions.” This statement should have adequate validation. The authors have not shown that PBMCs convert to ECFCs. Only have validated the presence of cells with an endothelial phenotype after isolating cells by density gradient.
Thank you for highlighting this, and apologies for the confusion. The PBMCs were in separate wells and one well differentiated into what we are now calling endothelial-like cells, and the other into blood outgrowth cells. The text in the results section has been updated to reflect this (Line 96).
Reviewer: The exposed methodology seems very confusing to me.
In my experience, the description of populations from PBMCs partially coincides with what we usually see. Between 4-7 days the first colonies appear, known as early-EPC, these cells usually always appear. Subsequently, after 15 and 20 days of culture (although only in a percentage of 10-20% of cases), colonies known as late-EPC (also known as ECFC) appear.
I still think that for this work to make sense, the authors should identify what type of endothelial cell they are using. Mainly, if it is ECFC, since these cells have been widely described in the literature and are of great interest to the scientific community. Above all, if these cells are correctly isolated from animal models. The different drug agencies state in their reports the need to use preclinical experimental animal models before undertaking clinical trials. A pig model, in which we can validate the potential of ECFCs as reconstruction therapies or tissue engineering, could be a magnificent tool.
- Line 250: “vWF staining was higher in the ECFCs than PAECs, and mRNA expression confirmed this finding. There was a large amount of variability between mRNA expression in ECFC, as highlighted by the large SEMs. This is in keeping with previous studies and is thought to be a reflection of maximum cell density [47]”.
I believe that this is associated with the heterogeneity of the different populations of ECFCs that are purified from blood. To avoid this heterogeneity, the authors could separate the populations by groups with the same clonal ability and would have more homogeneous cell populations.
We thank you for noting this valid point, and this is definitely something that could be investigated further in future studies. The manuscript text has been modified highlighting this (Line 279-282).
- Line 26: “we confirmed the feasibility of seeding derived porcine ECFC onto a decellularised human vein scaffold with approximately 90% lumen coverage at lower passages, and show that increasing cell passage results in reduced endothelial coverage”. The authors must show images that allow us to visualize the new endothelial cells lining the mammary artery.
We agree this suggestion will add weight to the manuscript. Figure 4 (Line 162) and it’s corresponding legend (Line 163) have now been updated to also include images of serial sections of seeded human saphenous vein, stained with DBA-lectin for endothelial-like cells lining the lumen, CD31, and H&E for cell nuclei. The methods have been updated to include CD31 immunohistochemistry (Line 507-510), and a line of text added to results to reflect this (Line 157).
Reviewer: This figure improves the quality of work. However, since the authors use pig cells in the pig, this makes it difficult to distinguish whether or not the protocol actually worked.
To avoid this, the authors could address different scenarios:
1- Show a negative control. It could be the decellularized vein. In this image, hardly any marked endothelial cells should be seen,
2- Mark the endothelial cells to be able to follow them. This technique would be the most visual and safe, in the image only the added cells would appear and not the remains of endothelial cells that may have remained after Decellularization.
Author Response
Additional Rebuttal to reviewer 2:
Generic reviewer comments and changes made: We again thank the reviewer for their constructive and useful comments on this manuscript, and we are grateful that the reviewer has appreciated most of the changes we have made following the initial round of revision. However, the reviewer appears to have a couple of residual concerns related to 1) cell nomenclature, suggesting that cell surface markers alone cannot categorically confirm that they are ECFCs and 2) experiments to demonstrate the clonal ability of the cells as a way to reinforce the suggestion that they are indeed ECFCs. We are grateful for these additional suggestions, which we have addressed in this further revision as follows:
- We have kept the term endothelial-like cells (ELC) in the manuscript, as a way to concede on the point that we cannot be 100% sure based on cell surface markers alone that these are ECFCs, and have now revised our text to highlight better this aspect in the manuscript to the reader (see Line 188-189).
- Regarding confirmation of clonal ability, unfortunately we are unable to undertake this experiment at this stage as we have used all the available cells for seeding on acellular biological vascular scaffolds implanted in vivo as part of an ongoing experiment. Given the apparent ease of the clonal ability assays, we will confirm this in future cell isolations to complete cell characterisation and to establish we have true ECFCs. However, we have now added in the revised paper phase contrast images that we obtained during culture of colonies formed by the cells when passaged (see Figure A1, Line 553). While this is not a test of clonal ability, this does confirm that once plated they still grow as colonies. We have also amended the text to cover this additional finding (see Line 183). We feel that this further enhances our current approach of naming these cells ELC, adding on the characterisation of the cell type.
Specific comments by reviewer and point-to-point reply: We have revised our manuscript taking the reviewers specific comments into consideration, and the responses to each point are below including the details of any changes made to the manuscript text. The manuscript sections referred to in the responses below have also been highlighted in the corrected manuscript file attached.
1. ……. I still think that the different approaches followed may have compromised the results. This would imply that their conclusions could be wrong. I think that the authors should simplify the confounding variables.
We thank the reviewer for their comments, but we believe there are differences between pig and human blood in relation to ELC isolation and growth (as previously outlined in the discussion from Line 195). As we have previously discussed, the isolation protocols for human ECFCs have been well defined, whereas the isolation from pig blood, as a highly relevant preclinical model for translational vascular tissue engineering, has been less well documented (Line 72). Hence, we believe that reporting all of the approaches attempted for isolation and expansion of porcine cells could be of great value to those readers working in this area. Indeed, showing the multiple approaches and by learning from our developmental work also aims to highlight potential issues, and helps streamline their approach to porcine ELC isolation e.g. only one type of specific manufacturers media appears to result in successful porcine ELC isolation, a key insight that that could ensure researchers do not unnecessarily waste animals, time and money.
2. Proper identification (nomenclature) is essential, among other reasons, so that other authors can subsequently reproduce the results observed in this study. The lack of proper identification has been one of the main problems in the study of endothelial cells and their potential in recent years. Different authors have spent years demanding an adequate characterization of these cells with therapeutic potential (Stem Cells Transl Med. 2017 May; 6(5):1316-1320. doi: 10.1002/sctm.16-0360). Any endothelial cell (progenitor or mature) and even other cells with phenotypes similar to those of endothelial cells (EPC, CAC,...), have the ability to adhere to different substrates, including the walls of arteries and veins. In this sense, in addition to a correct identification (which is not limited to surface markers), it is necessary to carry out additional tests that demonstrate that these cells are putative endothelial cells and that after purification and expansion it’s remain functional. There are many tests that validate the functionality of endothelial cells, some of the most used are tests such as Proliferation; Migration in response to VEGF-A and FGF-2; Angiogenesis assays (representative micrograph of endothelial cells-lined capillary-like network on a Matrigel coated plate); Responsiveness to inflammatory stimuli (Up-regulation of E-selectin and ICAM-1 in response to TNF-a); etc.
We thank the reviewer for these general remarks around the research area, however we don’t believe they are strictly relevant to our manuscript. Among other things, this highlights the need for a proper cell nomenclature and for validation of endothelial functionality text in vitro. We assume that this is related to work done on human cells as we know that very little has been done on porcine blood derived ELCs. We have clearly provided a set of preliminary experiments focusing on development and preliminary testing in vitro and in vivo on how to derive and expand these porcine ELCs, providing tips and pitfalls and a new avenue for our group, and others, to expand on the characterisation in future work. We believe that this is a strong start and see our preliminary findings helping ourselves and others to expand and complete the characterisation of these cells going forward.
However, I think this work would make sense if the authors could show that they have purified ECFCs. If the authors have ECFC, they should additionally prove it with clonal ability tests. If not, they should perform additional tests, to show that they are not using other endothelial-like cells or even mature endothelial cells whose expansion could be compromising their functionality. By definition, an ECFC is characterized by its clonogenic ability, large expansion potential, stable endothelial phenotype, and robust vessel-forming capacity. These are in concordance with the set of cell criteria that, ideally, should follow future vascular cell therapies and tissue engineering applications: (i) easy to isolate and purify; (ii) large proliferative potential; and (iii) robust vessel-forming ability [Tissue Eng Part A. 2009;15:3473-3486].
Thank you for these comments, and as mentioned in our opening statement above, we have added a figure containing cells forming colonies and comment on the nomenclature issue at Line 188 of the manuscript.
3. If the purified cells are ECFC, clonal ability tests are very simple and inexpensive. They only require the medium that was used for their purification and expansion. Colony formation is very easy to quantify. There are different protocols in the literature.
Please see our opening comment in this rebuttal (page 1, paragraph 1).
4. As I have commented in the previous section, the authors must better define what type of cells they are using. Many types of cells identified as endothelial cells have been identified in the blood. Among other reasons, the great variety of nomenclatures often makes reproducibility unfeasible.
Please see our opening comment in this rebuttal (page 1, paragraph 1).
5. I don't understand this answer. If the authors seeded the PBMCs in culture plates (without additional separations), how could different populations appear in different wells? In my experience when I have purified EPC or ECFC from PBMCs, both populations appear in the same culture dish, but at different times.
We apologise that we did not clarify this sufficiently in our previous response to the Reviewer’s comment. As shown by the ICC staining of blood outgrowth cells (BOC) (Figure 2b-d), there is the possibility that endothelial-like cells (ELC) were present in the BOC cultures, however due to the speed BOC emerged and proliferated in culture, it is unlikely that the BOC were present in wells of ELC that presented with a cobblestone-like morphology (this has now been added to Line 333). This might suggest that there are very few endothelial cell progenitor cells in the circulation as mentioned in the manuscript from Line 195. If all adhered monocytes resulted in ECFC, then we would agree that it wasn’t possible to get heterogenous cell type populations in different wells. However due to the low numbers of circulating progenitors, it may be possible by chance to have individual pre-cursors for different cell types in different wells, that go on to form different cell populations or types under the right conditions. For example, it has been shown that it is possible to differentiate human PBMCS into smooth muscle cells (Ahmetaj-Shala et al, 2021 doi.org/10.3389/fcell.2021.681347). Please note this reference has now been included at Line 342 to highlight this.
6. The exposed methodology seems very confusing to me. In my experience, the description of populations from PBMCs partially coincides with what we usually see. Between 4-7 days the first colonies appear, known as early-EPC, these cells usually always appear. Subsequently, after 15 and 20 days of culture (although only in a percentage of 10-20% of cases), colonies known as late-EPC (also known as ECFC) appear. I still think that for this work to make sense, the authors should identify what type of endothelial cell they are using. Mainly, if it is ECFC, since these cells have been widely described in the literature and are of great interest to the scientific community. Above all, if these cells are correctly isolated from animal models. The different drug agencies state in their reports the need to use preclinical experimental animal models before undertaking clinical trials. A pig model, in which we can validate the potential of ECFCs as reconstruction therapies or tissue engineering, could be a magnificent tool.
We thank the reviewer for this comment, and we are pleased to see our description partially coincides with the Reviewer’s experience. We highlight here again that we have worked on fresh porcine blood and that very little is reported in the literature on this large mammalian blood source in relation to deriving ELCs. We are pleased that this reviewer shares our enthusiasm that further validation of our work “could be a magnificent tool”. Please see our previous response to comment 5 regarding different cell populations, and our opening generic comment regarding identification of cells (page 1, paragraph 1).
7. This figure improves the quality of work. However, since the authors use pig cells in the pig, this makes it difficult to distinguish whether or not the protocol actually worked.
To avoid this, the authors could address different scenarios:
1- Show a negative control. It could be the decellularized vein. In this image, hardly any marked endothelial cells should be seen,
2- Mark the endothelial cells to be able to follow them. This technique would be the most visual and safe, in the image only the added cells would appear and not the remains of endothelial cells that may have remained after Decellularization.
We agree that a negative control would improve this figure, and as such have added Figure 4f (Line 163) and updated the Figure legend (Line 168) accordingly, to show a cross-section of DBA-lectin stained decellularised human saphenous vein as a negative control. In this instance, pig cells were seeded onto portions of decellularised human vein, therefore cell tracking/marking is not necessary in this instance. Moreover, it is important to note that DBA-lectin does not bind to human endothelial cells as it specifically binds to porcine endothelial cells. If these seeded vessels were implanted into pigs, then we agree that cell tracking would be helpful.
